# Pathogen Reduction for Platelets—A Review of Recent Implementation Strategies

**DOI:** 10.3390/pathogens11020142

**Published:** 2022-01-24

**Authors:** Paolo Rebulla, Daniele Prati

**Affiliations:** Department of Transfusion Medicine and Hematology, Foundation IRCCS Ca’ Granda Ospedale Maggiore Policlinico, 20122 Milan, Italy; daniele.prati@policlinico.mi.it

**Keywords:** platelets, pathogen reduction, platelet transfusion, transmissible infections

## Abstract

The development of pathogen reduction technologies (PRT) for labile blood components is a long-pursued goal in transfusion medicine. While PRT for red blood cells and whole blood are still in an early phase of development, different PRT platforms for plasma and platelets are commercially available and routinely used in several countries. This review describes complementary strategies recommended by the US FDA to mitigate the risk of septic reactions in platelet recipients, including PRT and large-volume delayed sampling, and summarizes the main findings of recent reports discussing economical and organizational issues of platelet PRT implementation. Sophisticated mathematical analytical models are available to determine the impact of PRT on platelet costs, shortages and outdates in different settings. PRT implementation requires careful planning to ensure the availability of sufficient economical, technological and human resources. A phased approach was used in most PRT implementation programs, starting with adult and pediatric immunocompromised patients at higher risk of developing septic platelet transfusion reactions. Overall, the reviewed studies show that significant progress has been made in this area, although additional efforts will be necessary to reduce the storage lesion of PRT platelets and to expand the sustainable applicability of PRT to all labile blood components.

## 1. Introduction

The development and general use of pathogen reduction technologies (PRT) for all labile blood components is still a well-recognized and long-pursued goal in transfusion medicine. The interest in these technological developments has risen in the last few decades to protect transfusion recipients not only from pathogens for which blood donors are regularly screened but also from those that may be undetected due to testing failures or blood donation during the infection window phase of classical transmissible viruses (e.g., HIV, HBV, HCV), and from emerging unknown or unscreened pathogens. The recent COVID-19 pandemic, although there have been no reports of transmission of its causative agent SARS-CoV-2 through blood transfusion, offers an opportunity to discuss our preparedness to handle emerging cases of novel transmissible agents [1].

Technical progress in this area has been conditioned by the differential levels of frailty of plasma proteins (particularly the labile blood coagulation factors and inhibitors) versus the cellular blood components (platelet and red blood cells) and by the need to preserve their clinical efficacy while avoiding unacceptable toxicity to the recipient. Specific approaches to PRT have been developed for plasma, platelets and red blood cells or whole blood in consideration of the different biochemical makeup of these complex and fragile biological materials.

As compared to blood components stored at 2–6 °C, earlier PRT programs have been developed for platelets due to the recognition of an increased risk of bacterial contamination associated with their storage at 20–24 °C. While PRT for red blood cells and whole blood are still in an early phase of development or experimental clinical use, different PRT platforms for plasma and platelets have already reached an advanced stage of regulatory approval, commercial availability and routine or selective use in several countries. Similar to other expensive procedures implemented to reduce the risk of transfusion-transmitted pathogens, PRT implementation requires careful analysis of their cost/utility ratio in specific jurisdictions. A systematic literature review on the economic issues was published by LaFontaine et al. in 2021 [2].

The aim of this article is to review the main findings of recent reports discussing organizational issues related to the implementation of commercial platelet PRT for adult and pediatric recipients in different countries. The review will include short summaries describing (a) the available platelet PRT platforms, (b) the storage lesion associated with PRT, (c) the clinical safety and efficacy of PRT platelets and (d) risk mitigation strategies alternative or complementary to PRT.

## 2. Platforms for Platelet PRT

The currently available commercial platforms for platelet PRT include the Intercept Blood System (Cerus Corporation, Concord, CA, USA), the Mirasol PRT System (Terumo BCT, Lakewood, CO, USA) and the Theraflex UV-C Platelets System (Macopharma, Mouvaux, France). The systems are based on UV illumination of the platelet container with UV light of different wavelengths by using an illumination device delivering controlled doses of energy, in the absence (Theraflex UV-C Platelets System) or in the presence of photoactive chemicals, amotosalen and riboflavin in the Intercept Blood System and in the Mirasol PRT system, respectively. The Intercept Blood System procedure includes a step of post-illumination adsorption to reduce residual amotosalen before platelet storage.

All systems induce irreversible damage of nucleic acids, thus preventing their replication. Although the acronym PRT specifically refers to *pathogen* inactivation, the nucleic acid damage induced in viable leukocytes present in platelets prevents also the risk of transfusion-associated graft-versus-host disease, making gamma irradiation unnecessary for this indication. Detailed data on the levels of inactivation of viruses, bacteria, parasites and leukocytes achieved with the different platforms have been reported in the literature [3,4,5,6,7,8,9,10,11,12]. Additional technical information can be downloaded from the PRT manufacturers’ websites. Transfusion medicine specialists interested in the implementation of platelet PRT should contact their country’s competent authority to determine the local regulatory status of the different platforms when used for apheresis and whole-blood-derived platelets. Quality requirements for PRT platelets have been established by the US FDA [13] and by the European Directorate for the Quality of Medicines [14].

## 3. Storage Lesion, Clinical Safety and Efficacy of PRT Platelets

An expert interlaboratory consensus on the proteome changes in PRT platelets published in 2014 reported the main damages at the protein level detected with the Intercept, Mirasol and Theraflex systems [15]. The authors noted that the different PRT systems induce different protein damages related to the platelet storage lesion. In this regard, the consensus indicated that “Mirasol impacts adhesion and platelet shape change, whereas Intercept seems to impact proteins of intracellular platelet activation pathways. Theraflex influences platelet shape change and aggregation, but the data reported to date are limited” [15]. With specific reference to the Mirasol and Intercept systems, the proteomic approaches “demonstrate that Mirasol tends to affect more the cytoskeleton organization than Intercept treatment, which, in turn, affects primarily the aggregation pathway” [15]. Overall, the authors of the 2014 consensus concluded that the number of proteins damaged with the different PRT systems is relatively small, although the damages are capable of inducing an acceleration of the platelet storage lesion. This observation will be particularly important in relation to the possibility of extending the shelf life of PRT platelets.

Following the 2014 consensus, additional data on the impact of PRT on the platelet storage lesion were reported by Schubert et al. [16], who showed the occurrence of protein translation in PRT platelets treated with the Mirasol system, suggesting that “platelets may possess a mechanism to protect their mRNA from damage by the PRT treatment”. Very recently, new observations were reported by Sonego et al. [17], who investigated the impact of Intercept on redox cysteines and showed that “Intercept oxidized integrin bIII, which could activate the mitogen-activated protein kinases pathway”.

The above studies have disclosed novel mechanisms of platelet modifications induced by PRT that complement previous reports based on classical measures of the platelet storage lesion, including pH, morphology, hypotonic shock response, aggregation, glucose consumption and lactate formation, flow cytometry detection of markers of activation and post-transfusion survival. It can be expected that the new information provided with the proteome approaches could improve refinements of the currently available PRT systems and possibly increase the clinical efficacy of PRT platelets.

The high safety profile of PRT platelets is supported by the data published by different hemovigilance programs, which reported no cases of transfusion-transmitted infections in transfusion recipients [18,19,20].

The clinical efficacy of PRT platelets has been evaluated in seven, three and one randomized clinical trials using the Intercept, Mirasol and Theraflex systems, respectively. Although the in vitro studies [15,16,17] consistently documented the deleterious effects of PRT on the platelet storage lesion and the clinical trials showed decreased post-transfusion platelet count increments in recipients, the current evidence indicates that PRT platelets offer similar protection against clinically relevant bleeding as compared to conventional platelets [1,21,22].

## 4. Risk Mitigation Strategies Alternative or Complementary to PRT

This review on PRT platelets would be incomplete without considering the broader framework of other technologies that have been developed to reduce the risk of bacterial infection transmission through platelet transfusion. FDA guidance for industry [13] provides details for the implementation of three recommended risk mitigation strategies in the US: (a) large-volume delayed sampling (LVDS) of the platelet unit and use of a culture system started no sooner than 36 h post-collection, followed by a holding time of 12 h before release, with a platelet shelf life of 5 days, which can be extended to 7 days after performance of secondary testing (LVDS5/2); (b) LVDS with a 7-day platelet shelf life and use of a culture system started no sooner than 48 h post-collection, followed by a holding time of 12 h before release (LVDS7); (c) the Intercept PRT with a platelet shelf life of 5 days.

An interesting study funded and directed by bioMérieux, the manufacturer of a cell culture system approved for use with the LVDS strategies, evaluated the clinical and economic impact of the above three platelet processing strategies applied to the treatment of onco-hematology patients [23]. The authors used a decision analytical model with inputs derived from Medicare reimbursement costs and published literature on platelet requirements, recipients’ clinical outcomes and adverse reactions. Based on the above model, the authors concluded that the cost of using LVDS platelets was 55–58% of the cost of Intercept PRT platelets and that the LVDS strategies were associated with higher platelet availability as compared with the Intercept PRT processing strategy. It would be interesting to expand the findings of this study with additional industry-independent comparisons of the LVDS strategies versus the different commercially available PRT, including the option for 7 day storage of PRT platelets and a larger case mix of recipients. Moreover, further investigations are needed to determine if the outcomes of the decision analytical model developed by Earnshaw et al. [23] are confirmed in real-world settings.

Another decision analysis model was recently used by Helander et al. [24] to evaluate whether an inventory based on PRT platelets only is financially feasible in “a level 1 trauma center which does not have an in-house donor center and is reliant on external blood suppliers”. The study was carried out following an announcement in July 2020 by the American Red Cross on their plans to comply with the FDA guidance [13], which included an initial inventory conversion to 5-day LVDS platelets and a long-term plan to transition to a PRT platelets only inventory. The authors determined that their current cost of USD 5.70 million for an annual inventory of approximately 8550 leukoreduced apheresis platelets including 40% standard and 60% PRT platelets would increase to USD 5.99 (+5.1%) and 6.15 (+7.9%) million for an inventory of 40% LVDS and 60% PRT platelets and of PRT platelets only, respectively.

The above models, when used with solid local input data, can prove useful to estimate the economic impact of alternative or complementary risk mitigation strategies in different settings and geographical jurisdictions.

## 5. Implementation of PRT Platelets

A number of recent reports describe platelet PRT implementation procedures and outcomes in various settings and countries [25,26,27,28,29,30,31,32,33,34]. The presence of a mixed platelet inventory during a transition phase to PRT platelet implementation offers the opportunity to perform studies comparing the latter with conventional platelets.

### 5.1. Models for Inventory Management

Platelet inventory management, with a particular focus on shortages and outdates, was the main issue of two studies carried out with mathematical models in Spain [25] and Canada [26].

Gorria et al. [25] determined the reduction in platelet outdates associated with an extension of PRT platelets’ shelf life from 5 to 7 days with a mathematical simulation model using input data derived from the daily demand for platelets in 2016 at two blood centers in Spain producing yearly 11,500 (Basque Center for Transfusion and Human Tissues) and 6200 (Aragon Blood and Tissue Bank) platelet units. The study made no distinction between ABO and Rh groups and between apheresis and pooled platelets. Due to limitations caused by the number of available donations every day, the model included a maximum production limit set at 30 platelet units per day in both centers. PRT was carried out Monday to Friday at the Basque Center and Tuesday to Saturday at the Aragon Blood Bank. Depending on the different operational scenarios of the two blood centers, the authors found that an extension of platelets’ shelf life from 5 to 7 days would decrease outdates from 1.41–4.54% to 0.00–0.27%, respectively. Moreover, the authors noted that, in some settings, budgetary reasons could prevent PRT treatment for the entire platelet production process and concluded that if the proportion of PRT platelets exceeds 25%, “the best option is to treat part of the output every day, otherwise, it is preferable to concentrate treatment on the last two production days of the week” [25].

The Center Inventory Application (CIA), a custom-built simulation model, was used by Blake et al. [26] to estimate the impact on the inventory, wastage and shortages of implementing the Intercept platelet PRT in Canada as applied to pooled platelet concentrates prepared with the buffy-coat method (BCPC) and to apheresis platelets with a shelf life of 5 and 7 days, respectively. Specifically, wastage and shortage were determined “when a 5-day PRT-BCPC product was introduced alongside a 7-day apheresis platelet product” [26]. The CIA model used 2019 input operational data derived from approximately 12,000 transaction records from a pilot region belonging to the Canadian Blood Services, which processes approximately 8% of the total national platelet demand. The authors noted that “when the exigencies of outbound platelets distribution, which dispatches regular orders once per day, are included, the effective shelf life of BCPC will be either 3 or 4 days following the implementation of PRT. Since perishable products are subject to waste and shortage, the reduction in shelf life for PRT platelets will impact blood agency operations” [26]. In contrast with the model developed by Gorria et al. [25], which showed a decrease in outdates in a setting in which PRT allows an extension of platelets’ shelf life, Blake et al. [26] considered the effect of “a decrease in shelf life for platelets or the operational impact of managing a dual product inventory with differing shelf lives and product substitutability”. After validating the CIA model, Blake et al. [26] investigated three different scenarios. Scenario 1 simulated a comparison of a 5- to 3–4-day effective shelf life of PRT BCPC, which showed that reductions in PRT platelets’ shelf life resulted in shortages of both BCPC and apheresis platelets; scenario 2 determined the increase in BCPC and apheresis platelets needed to maintain the original (2019) service levels after PRT implementation; scenario 3 was based on the expectation that PRT implementation could determine a 20% increase in platelet demand due to increased per-patient transfusion requirements associated with lower post-transfusion platelet count increments. The authors concluded that PRT BCPC implementation would require 9% and 6% increments in BCPC and apheresis platelet production, respectively, “to maintain a non-inferior level of customer service while minimizing wastage (expected value of 15.8%)” [26]. An additional interesting observation collected with this model was that “while shortages increase as the shelf life of BCPC decreases, apheresis waste declines as BCPC shelf life decreases because apheresis units can be substituted for BCPC units” [26].

Besides the interest in the outputs of the mathematical models, different conclusions could be reached in settings with different operational conditions (such as internal platelet production versus provision by an external supplier), platelet demand and recipient case mix.

### 5.2. Phased PRT Implementation

Three studies describe the rationale and the outcomes of phased implementations of PRT platelets in operative settings using a mixed inventory of PRT and conventional platelets [27,28,29]. All studies noted the paramount importance of close cooperation between the manufacturer of the PRT system, the platelet providers (blood centers) and the platelet users (blood banks and hospitals) to ensure the smooth continuation of efficient service to platelet recipients during the transition phase.

Allen et al. [27] implemented PRT platelets in a university health system serving two hospitals with 750 beds for adult and neonatal inpatients and an outpatient cancer center with an annual demand of 9500 apheresis platelets. Approval and funding from the hospital quality council and administration was obtained 1 year before the planned change. A pre-implementation program was developed to let the laboratory information system manage PRT platelets and recognize their equivalence to leukoreduced, CMV-negative and gamma-irradiated platelets. Meanwhile, live, small group training sessions on the mechanism and efficacy of PRT were provided to the laboratory and clinical staff. Due to ethical and pragmatic reasons, the PRT platelet implementation program was phased by recipient population, with an original target to provide more than 90% PRT platelets within 6 months. In phase 1, started on 10 February 2017 and lasting 6.5 weeks, the target recipient population was limited to the cancer center outpatients, as these immunocompromised patients were at the highest risk of developing septic transfusion reactions and could have limited access to immediate advanced critical care after discharge. In phase 2, started on 27 March 2017 and lasting 2 weeks, inpatients from the bone marrow transplant ward, another severely immunocompromised patient group, were added. In phase 3, started on 10 April 2017, PRT platelets could be used throughout the health system. At 6 months, only 41% of all platelets were treated with PRT. A final phase 4 began in February 2018 (month 13) and the authors reported that 92% of all transfused platelets were PRT-treated in month 23 (December 2018). An investigation of the causes preventing the achievement of the original target of >90% PRT platelets within 6 months of implementation showed that the fraction of PRT platelets increased progressively with optimization strategies at the provider blood center, which required more time than originally estimated, and with the expansion of other facilities accepting PRT platelets.

Nguyen et al. [28] reported their phased Intercept PRT implementation in a setting including a hospital-based blood donor center, a blood component laboratory and a transfusion service transfusing approximately 13,000 apheresis platelets each year in an 800-bed two-hospital system and outpatient oncology centers. The authors described a pre-implementation process from the perspectives of the donor center, the component processing laboratory, the transfusion service, the clinical and nursing staff and the patients. The main issue at the donor center was to refine the settings of the Trima (Terumo Medical, Tokyo, Japan) apheresis device to ensure that apheresis collections would meet the Intercept input specifications (guard bands). This was achieved after the development of two and four novel settings for single and double collections, which progressively improved the platelet yield from 4.1 to 4.2 and from 7.0 to 7.4 × 10^11^, respectively. The component processing laboratory set four milestones: facility and process assessment, focusing on space and equipment, product qualification, equipment (two illuminators) and workflow validation, with the aim of maintaining the fraction of apheresis collections treated with PRT at approximately 50–55% of the total during the implementation phases. Staff training was started with the selection and training of ‘subject matter experts’, who were then tasked with conducting the second phase of training of the rest of the laboratory staff. Before and after PRT implementation, the transfusion service maintained their usual dual inventory of 70–75% and 25–30% units provided by the blood donor center and purchased by an outside vendor, respectively. Similar to Allen et al. [27], this program also required modifications of the laboratory information system allowing the recognition of the equivalence of PRT to irradiated and CMV-negative platelets. Grand rounds and staff meetings were organized to provide education on PRT, with a more intensive schedule for wards regularly transfusing platelets. Specific focus was placed on highlighting visual differences between non-PRT and PRT platelets, namely the bag size and the lack of CMV status and irradiation label on the latter. When both PRT and conventional platelets were available in the inventory, recipients from the neonatal, pediatric and adult intensive care units, onco-hematology wards and outpatient infusion centers were selected to receive PRT platelets. During phase 1, lasting 17 months, approximately 44% and 56% of apheresis collections were transfused as conventional and PRT platelets, respectively, with a discard rate below 2%. In phase 2, a software platform developed by Cerus was used to increase the number of units meeting the guard bands required for PRT by two pre-processing steps: “dose–volume mitigation” (DVM) and “splitting high doubles” (SHD). Use of DVM and SHD increased the percentage of PRT platelets from 56% to approximately 78%. Only two inquiries/concerns were received from clinicians and patients during the above implementation phases. While the authors concluded that “a phased implementation and maintenance of a dual inventory provides flexibility to blood collection, blood manufacturing, and transfusion service processes”, they also acknowledged a number of challenges identified during PRT implementation, including the requirement of one additional staff member and the need for PRT process revalidation “each time an optimization planning was implemented” [28].

A more recent study identified the need to comply with guard bands as “the most notable obstacle to efficient PRT platelet production” [29]. To overcome this limitation, the authors investigated three collection settings with distinct unit volume and platelet yield targets and determined the proportion of units eligible for PRT, the split rate, the proportion of overconcentrated and under-yield units. The setting with a target volume of 410 mL and yield of 6.8 × 10^11^ platelets “gave the most favourable results in terms of overconcentration, with only 3% of products needing to be discarded. However, this setting also resulted in 18.7% of platelet units being low yield, having lower than 3 × 10^11^ platelets per unit” [29]. In early March 2018, a new code was obtained for these low-yield PRT platelets, which contained on average 2.8 × 10^11^ platelets. PRT platelet production started in January 2018, with an initial target to distribute 20% PRT platelets during the first year. This percentage increased to 40–50% by the beginning of 2021. In this program, PRT platelets were also considered CMV-safe and equivalent to gamma-irradiated platelets. They were preferentially selected for immunocompromised patients if no other conventional platelet unit was to expire sooner. After a final optimization with the use of DVM and SHD, the program was expanded to a 100% PRT platelet supply in April 2021. The hospital and the Stanford Blood Center made a joint decision “to absorb the added cost of producing PRT platelets without an upcharge to patients due to the balancing effects of reduced bacterial culturing” [29].

A comparative analysis of platelet production, inventory management, discard rates, blood utilization and clinical outcomes over 40 months before and after a single-phase Intercept PRT implementation was reported by Fachini et al. [30] from the Blood Bank of the Sirio-Libanés Hospital in Sao Paulo, Brazil. A total of 7777 and 6921 platelet doses (each defined as 3 × 10^11^ platelets, 86.1% and 86.9% prepared by apheresis, 13.9% and 13.1% prepared from whole blood units) were transfused in the 40 months before and after PRT implementation in March 2017, respectively. The authors reported that in the post-PRT period, an increase in double and low-volume apheresis collections allowed 80.2% of all apheresis units to be treated with the Intercept double-storage disposable sets. Discard rates decreased from 6% to 3% after PRT implementation and to 1.2% after the extension of the PRT platelet shelf life from 5 to 7 days, approved by the National Regulatory Agency (ANVISA) in June 2020. Platelet utilization was comparable before and after PRT implementation, with a mean number of 5.86 and 5.56 units/patient, respectively (*p* = 0.5787). The reported adverse events decreased from 2.15% before to 1.41% after implementation, a difference mainly driven by a reduction in mild allergic reactions from 1.63% to 1.11%, respectively.

### 5.3. Pediatric and Adult Recipients

Investigators led by Edward L. Snyder from Yale University described blood utilization and transfusion reactions in two retrospective studies carried out in pediatric [31] and adult [32] platelet recipients during a transition phase from conventional to Intercept PRT platelets, started in November 2016. Conventional and PRT platelets used in these studies were manufactured by the American Red Cross or the Rhode Island Blood Center. The conventional platelets were tested with bacterial cultures at the collecting blood centers and further analyzed at the authors’ blood bank on storage day 4 or 5 with the Platelet PGD bacterial mitigation assay (Verax Biomedical, Marlborough, MA, USA).

The pediatric recipients studied by Schulz et al. [31] over a 21-month period during the transition from conventional to PRT platelets included 72, 45 and 131 neonatal intensive care unit patients (NICU), 0–12-month-old infants not admitted to NICU (INF) and 1–18-year-old children (PED), respectively. The NICU, INF and PED patients received 91 and 145, 125 and 254, 644 and 673 conventional and PRT platelets, respectively. The proportion of PRT platelets transfused in the three groups during the study period ranged from 51% to 67% of the total. A statistically significant (*p* < 0.001) greater use of subsequent platelet doses was observed after PRT platelet transfusion (mean 1.4 ± 2.2) in the PED group as compared to conventional platelets (mean 0.9 ± 1.6), while differences were not statistically significant in the NICU and INF groups. To assess differences in red blood cell use as a proxy for clinically significant bleeding, the authors determined the number of red cell doses transfused within 48 h after platelet transfusion and found no statistically significant differences between PRT and conventional platelet recipients in all three groups. A total of 6/860 (0.7%) and 4/1072 (0.4%) allergic or febrile transfusion reactions were passively reported following PRT and conventional platelet transfusions, respectively, with a statistically non-significant difference. Of particular interest for NICU patients, the authors reported that “because of the FDA label for pathogen-reduced platelets including a caution for the potential development of skin rashes in neonates receiving psoralen compounds and phototherapy” [31], a neonatologist reviewed the NICU patients’ clinical records “for possible skin reactions in patients undergoing concomitant platelet transfusion and phototherapy” [31]. No rashes associated with PRT platelets were noted in this study, a finding that was expected by the authors “given that phototherapy devices approved for use in the US have a peak energy wavelength higher than the pathogen-reduced platelet label’s recommended wavelength cut-off of 425 nm, and nearly all have a lower-end emission higher than the 375 nm wavelength cut-off where an interaction with psoralen would be of concern” [31].

In the parallel study on adult recipients carried out by Bahar et al. [32], 3767 patients received 8912 conventional and 12,995 PRT platelets over a 28-month period. A proportion of patients received both conventional and PRT platelets, while 1087 received only conventional components (*n* = 1578) and 1466 were transfused with only PRT platelets (*n* = 2604). Consistent with previously published evidence [1,22], a statistically significant, slightly higher number of platelet transfusions was noted in the PRT only group (1.78/patient, 95% CI: 1.7–1.91) as compared to the conventional components only group (1.45/patient, 95% CI: 1.4–1.5). Conversely, the study showed a statistically significant, slightly lower number of red blood cell units transfused after PRT platelets as compared to conventional platelets. The authors reported no cases of septic reactions after the transfusion of the 12,995 PRT platelets, while five septic transfusion reactions occurred after the 8912 conventional platelet transfusions, “despite the use of point-of-release testing for two of these platelet units due to false negative results” [32]. No differences were found for other types of passively reported transfusion reactions.

A more recent study by Lasky et al. [33] reported a retrospective comparative analysis of the demographics, transfusion rates and transfusion reactions in 191 pediatric patients (<18 years old), including 51 under 4 months of age, who were transfused with at least one platelet unit during a 300-day period (February–December 2017), when a dual inventory of conventional and PRT apheresis platelets was available. Similar to the study by Schulz et al. [31], a manual search for evidence of rash was performed in the medical records of patients transfused with PRT platelets. Two thirds of the 892 platelet units analyzed during the study were given to onco-hematology recipients, with smaller proportions used for surgery patients (15.7%), thrombocytopenia in liver failure (4.5%), high-risk neonates (3.9%), bleeding (3.5%), disseminated intravascular coagulation (3.1%) and other less frequent conditions. Sixty-seven, 68 and 56 patients received both conventional and PRT platelets, PRT only platelets and conventional only platelets, respectively. Five allergic transfusion reactions were reported after the transfusion of 506 PRT platelets (0.99%), while one allergic and one transfusion-associated circulatory overload reaction occurred after the transfusion of 386 conventional platelets (0.52%). This difference was not statistically significant (*p* = 0.7052). In particular, no transfusion reaction was noted in the subgroup of 37 neonates who received PRT platelets, alone or in combination with conventional platelets. No transfusion transmitted infections were identified.

Finally, retrospective data from 1980 adult and 379 pediatric recipients (including 132 neonates) of Mirasol PRT platelets obtained with the buffy-coat method (45%) or by apheresis (55%), stored in 35% plasma and 65% platelet additive solution and transfused during 2013–2017, were reported by Jimenez-Marco et al. [34]. This study showed very low, not statistically significantly different frequencies of adverse reactions reported among the adult and the pediatric recipients (0.19% versus 0.12%, respectively, *p* = 0.85). Moreover, the authors performed a preliminary comparison of the PRT platelet requirements (*n* = 458) in the 132 neonates transfused during 2013–2017 with those of a historical cohort of 99 neonates transfused with 176 standard platelets stored in 100% plasma during 2003–2007. The two distant study periods were chosen “because in the years 2008–2012 the blood bank temporarily used an alternative PRT to treat platelets (not based on riboflavin and UV light)”. Although this analysis showed “a statistically significant increase in platelet usage in neonates transfused with PRT platelets”, the authors acknowledged the limitation associated with the use of data from control neonates transfused 10 years before with platelets stored in 100% plasma and concluded that “a long term follow up in chronically transfused pediatric patients and clinical trials on riboflavin and UV light-PRT and other PRT systems are needed to support the safe use of PRT-treated platelets in children”.

## 6. Review Limitations

This review has limitations. First, the discussed literature includes a small number of recent reports on PRT implementation. However, readers interested in additional information may find it in the references listed in the recent reports, which provide a full picture of previous implementation studies that have been so far performed. Second, most reports describe the successful implementation of PRT platelets in various settings among pediatric and adult recipients. Although this is encouraging, publication bias may have decreased the opportunity for the publication of less successful programs. Third, the implementation programs described in this review mostly refer to one PRT manufacturer; therefore, it should be noted that the conclusions reported here cannot be immediately generalized to the use of different platforms.

## 7. Conclusions

Several programs reported in the recent literature can be valuable models for a phased, smooth implementation of PRT platelets for adult and pediatric transfusion recipients. However, it should be appreciated that many variables can impact the efficacy of PRT implementation. Therefore, plans for each PRT program—alone or in combination with the use of LVDS—should be based on local, accurate input data including economic sustainability, epidemiology of transfusion transmitted infections, blood donor availability, platelet demand and recipient case mix.

A number of key messages derived from this review can facilitate the critical reading of the current and future literature on PRT platelet implementation.

Published evidence on platelet PRT implementation is limited for some platforms.

Data reported in the literature should be considered with attention to publication independence and publication bias.

The impact of PRT on transfusion-transmitted infections may vary in geographical areas with different prevalences of infectious agents.

Costs and benefits of platelet PRT should be estimated using validated models with input data from the local setting.

The impact of the “effective” versus the “regulatory” shelf life of conventional and PRT platelets should be considered in each operational setting.

Risk mitigation strategies alternative or complementary to PRT are available.

A phased transition from conventional to PRT platelets may help in the maintenance of an effective and efficient service.

## Data Availability

Not applicable.

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
