# Peer review of "Pathogen Reduction for Platelets—A Review of Recent Implementation Strategies"

_pathogens, 2022, doi:10.3390/pathogens11020142_

Round 1
Reviewer 1 Report
Valuable review.
The English needs minor review. A nice graphical abstract, expaining the message of the review in a clear way, would strengthen the manuscript.
Reviewer 2 Report
Dr Rebulla and colleague review the implementation strategies of pathogen reduction technology for platelets. Before, they summarize the PRT platforms available, the storage lesion that produce on platelets and their clinical safety. Since the implementation of PTR is a topic of great interest, this review provides critical information for those blood centers that could be interested in changing their platelet production. The article is well-written, the implementation strategies are well-described.
It can be considered with minor changes:
- Quality Requirements for PTR platelets should be commented (as those provided by European Committee).
- This reference should be introduced and commented because provides interesting data about the impact of PTR platelet transfusions on children:
Use and safety of riboflavin and UV light-treated platelet transfusions in children over a five-year period: focusing on neonates. Jimenez-Marco T, Garcia-Recio M, Girona-Llobera E. Transfusion. 2019;59:3580-3588.
Author Response
Referee 2. I appreciate the valuable comments. All text modifications in rev 1 are reported in yellow.
Comment 1. Quality Requirements for PTR platelets should be commented (as those provided by European
Committee).
Response 1. References 13 and 14 in rev 1 have been added to refer to FDA (US) and EDQM (EU) quality
requirements for PRT platelets (page 3, line 2 in rev 1).
Comment 2. This reference should be introduced and commented because provides interesting data about
the impact of PTR platelet transfusions on children: Use and safety of riboflavin and UV light-treated platelet
transfusions in children over a five-year period: focusing on neonates. Jimenez-Marco T, Garcia-Recio M,
Girona-Llobera E. Transfusion. 2019;59:3580-3588.
Response 2. This reference (34 in rev 1) has been added and the study has been commented (page 8, line
41)

Reviewer 3 Report
In the work (Manuscript ID pathogens-1554968d ) entitled ”Pathogen Reduction for Platelets: a review of recent implementation strategies” by Paolo Rebulla and Daniele Prati, the authors review the main findings of recent reports on pathogen reduction technologies (Prt) to platelets, including a) platforms for platelet PRT, b) Storage lesion, clinical safety and efficacy of PRT platelets., c) Risk mitigation strategies alternative or complementary to PRT and d) implementation of PRT platelets. I have the following comments:
Most of the text in the abstract is dedicated to introductory aspects, while what is obtained in the review is scarcely considered and superficially. It would be convenient to extend this last.
The limitations of the review could be exposed in a section prior to conclusions.
Table I can be kept in the conclusions, adding in each section, the references in this regard for those that are not proper. Therefore, in the conclusions, the explanation of the sections that correspond to the Table I and the limitations can be added in a brief way.
Author Response
Referee 3. I wish to thank the referee for the proposal to revise the abstract and to improve the format of the
final part of the review. All text modifications in rev 1 are reported in yellow.
Comment 1. Most of the text in the abstract is dedicated to introductory aspects, while what is obtained in
the review is scarcely considered and superficially. It would be convenient to extend this last.
Response 1. I have re-written the abstract reducing the introductory part and expanding the review’s
findings.
Comment 2. The limitations of the review could be exposed in a section prior to conclusions.
Response 2. Done (page 9, line 10).
Comment 3. Table I can be kept in the conclusions, adding in each section, the references in this regard for
those that are not proper. Therefore, in the conclusions, the explanation of the sections that correspond to
the Table I and the limitations can be added in a brief way.
Response 3. I appreciate the referee’s comment, but respectfully submit that I do not consider appropriate
to specifically list articles related to some of key messages as ‘not proper’. In fact, in my opinion those
publications are not ‘not proper’, they are just industrially sponsored. Therefore, we simply recommend that
they should be read considering this fact. I also changed the style of the final key messages (page 9, line 27)
from a table format to a list of bullet points, as I believe this is more stylistically appropriate for the
conclusions portion of the manuscript.
